# Retrospective Comparison of Anesthetic Methods for Percutaneous Balloon Kyphoplasty Surgery: General Anesthesia and Erector Spinae Plane Block

**DOI:** 10.3390/medicina59020240

**Published:** 2023-01-27

**Authors:** Ufuk Demir, Öztürk Taşkın

**Affiliations:** Department of Anesthesiology and Reanimation, Kastamonu University, 37150 Kastamonu, Turkey

**Keywords:** percutaneous balloon kyphoplasty, vertebral fracture, general anesthesia, erector spinae plane block, regional anesthesia

## Abstract

*Background and Objectives*: This study aims to investigate whether erector spinae plane (ESP) block can be an alternative to general anesthesia as a primary anesthesia method in percutaneous balloon kyphoplasty surgery. In addition, postoperative intensive care needs were compared in terms of length of intensive care unit stay and length of hospital stay. *Materials and Methods*: Medical records of patients who underwent percutaneous balloon kyphoplasty operation at Kastamonu Training and Research Hospital between January 2020 and November 2022 were reviewed retrospectively. Among 70 patients who underwent percutaneous balloon kyphoplasty, 58 patients with ASA (American Association of Anesthesiologists) classification III or IV, who underwent general anesthesia as the anesthesia method or ESP block, were included in the study. The patients were divided into two groups according to the anesthesia method applied. Group GA: general anesthesia group (*n* = 19) and Group ESP: ESP block group (*n* = 39). Group data for age, gender, ASA classification, number of surgical levels, surgical urgency, duration of surgery, postoperative complications, need for intensive care, length of stay in the intensive care unit, and hospital stay were compared. *Results*: There was no statistically significant difference between Group GA and Group ESP in terms of age, gender, ASA classification, surgical urgency, number of surgical levels, duration of surgery, and postoperative complication data of the patients included in the study. Seven (36.6%) patients in Group GA and six (15.4%) patients in Group ESP needed intensive care, and there was no statistically significant difference between the groups (*p* = 0.06). However, the need for intensive care was higher in Group GA. A statistically significant difference was found in Group GA and Group ESP, which was higher in Group GA for the length of stay in the intensive care unit (*p* = 0.02) and length of hospitalization (*p* = 0.04). *Conclusions*: ESP block may be an alternative method to general anesthesia as the primary anesthetic option for single or multilevel percutaneous balloon kyphoplasty surgery. With the ESP block, the length of stay in the intensive care unit and the length of stay in the hospital can be reduced, especially in high-risk patients.

## 1. Introduction

Vertebral compression fractures occur secondary to bone weakening from osteoporosis, metastatic disease, and other pathological processes [1]. In patients with vertebral compression fractures, pain may persist despite non-surgical treatments including analgesics, bed rest, physiotherapy, and back support and may cause balance disorders [1,2].

Percutaneous balloon kyphoplasty is one of the minimally invasive surgical techniques used to treat vertebral fractures. This technique provides good fracture stabilization and early reduction in postoperative pain. The incidence rates of surgical complications and side effects are low. The most common indication for balloon kyphoplasty is a painful compressive osteoporotic fracture of the vertebral body [1].

Percutaneous balloon kyphoplasty can be performed under local anesthesia, general anesthesia, regional anesthesia, sedation, or a combination of these, depending on the location and number of the vertebrae [1]. However, deep sedation in the prone position is risky regarding airway safety [3,4]. It can also be applied under general anesthesia, but general anesthesia may not be preferred because the patients who undergo kyphoplasty operations are usually risky patients with multiple comorbidities [5].

When applied alone or combined with mild sedation, regional anesthetic techniques significantly reduce pain intensity compared to local infiltration anesthesia alone. They have been reported to reduce side effects or postoperative complications associated with general anesthesia [4].

Regional anesthesia and pain management have made progress in recent years with the emergence of fascial plane blockades. The main advantages of these techniques include ease of application, analgesic efficacy, and low risk of postoperative complications. Among the regional anesthetic methods, epidural and paravertebral blocks are also available. Still, they are more invasive, challenging to apply, and have a higher complication risk than fascial plane blocks [6].

General anesthesia is a reversible amnesia state that allows patients to undergo surgical procedures safely and painlessly. Although it is increasingly safe, general anesthesia has risks and perioperative complications [7]. Complications associated with general anesthesia range from minor complications with no long-term consequences to complications with long-term repercussions that result in permanent disability. Cardiovascular and respiratory complications are the most common and are important determinants of postoperative morbidity and mortality. Perioperative cardiac complications include hypotension, myocardial ischemia or infarction, heart failure, and cardiac arrest. Perioperative respiratory complications related to anesthesia include desaturation, atelectasis, aspiration, and bronchospasm. Other serious complications include the development of acute renal failure and prolonged postoperative cognitive dysfunction. Minor but significant complications of general anesthesia include postoperative nausea and vomiting (PONV), sore throat, and tooth damage. All these perioperative complications can have a significant impact on patients and lead to the prolonged hospital stay, costs, and complications associated with prolonged hospitalization [8,9].

One of the newest techniques described recently is the erector spinae plane (ESP) block. The technique is easy to apply and has a low postoperative complication rate. After ultrasound-guided injection between the erector spinae muscle and the transverse process of the vertebra, the local anesthetic fluid diffuses cranially and caudally and acts on the ventral and dorsal branches of the spinal nerves [10]. They were first described by Forero et al. [11]. Since then, the ESP block has gained popularity in perioperative analgesia for a variety of surgical procedures such as breast, thoracic, abdominal, and lumbar spine surgeries [10]. In patients who underwent ESP block for analgesia in lumbar surgery, decreased opioid consumption, lower incidence of PONV, better patient satisfaction, and shorter hospital stay were reported [10]. In their meta-analysis, Ma et al. [12] showed that ESP block effectively reduced postoperative pain scores and postoperative opioid consumption in spine surgery. Ultrasound-guided ESP block is an interfascial plane block often used for postoperative analgesia. It has been reported that it is preferred as an anesthetic technique in kyphoplasty operations [3,5].

In this retrospective study, we aimed to investigate whether ESP block can be an alternative to general anesthesia as the primary anesthetic method for kyphoplasty operations. In addition, we aimed to compare ESP block and general anesthesia methods for kyphoplasty operation in terms of postoperative intensive care need, length of intensive care unit stay, and length of hospital stay.

## 2. Materials and Methods

This study was planned by the Helsinki Declaration decisions, the Patient Rights Regulation, and ethical rules. Ethical approval for this retrospective study was acquired from the Clinical Research Ethics Committee of Kastamonu University Faculty of Medicine, Kastamonu, Turkey (Approval no.: 2022-KAEK-127).

### 2.1. Data Collection

Demographic data of the patients, including age, gender, and ASA classification, were recorded from the hospital’s electronic medical record system. Personal data such as name, surname, and identity number were not recorded. In addition, the type of anesthesia administered during the intraoperative period, the need for intraoperative rescue analgesia, whether the operation could be completed with the current anesthesia method, the vertebral level at which percutaneous balloon kyphoplasty was applied, the number of vertebrae that underwent surgery, and the duration of the operation were recorded from the medical records. Postoperative data, complications developed after the operation, the need for intensive care, length of stay in the intensive care unit, and length of hospitalization were recorded. The length of stay in the intensive care unit and the length of stay in the hospital given in our study refer to the postoperative period.

### 2.2. Patient Population

Medical records of patients who underwent percutaneous balloon kyphoplasty operations in Kastamonu Training and Research Hospital operating rooms were reviewed retrospectively between January 2020 and November 2022. Among these patients, patients with ASA (American Society of Anesthesiologists) classification III or IV, who underwent general anesthesia as a method of anesthesia or underwent ESP block, were included. In our study, the exclusion criteria were patients with ASA classification I and II who underwent another operation at the same time as percutaneous balloon kyphoplasty surgery and were administered sedo-analgesia alone as an anesthesia option, and who lacked data in the medical records. In addition, additional inclusion and exclusion criteria were applied to ensure standardization in the anesthesia and surgical methods.

According to descriptive statistics (effect size = 1.025) in the study by Ge C et al. [4] and two-sample *t*-test power analysis, a total sample size of 47 was found to achieve 80% power with the classical 0.05 significance level (Hintze, J. (2011). PASS 11. NCSS, LLC., Kaysville, UT, USA. https://www.ncss.com/support/faq/ (accessed on 15 January 2023)). Since our study was a retrospective study, all patients who met the inclusion and exclusion criteria were included and our study was therefore completed with 58 patients.

#### 2.2.1. Anesthesia Method and Patient Selection

There were no additional criteria for inclusion for the general anesthesia group except endotracheal intubation with standard general anesthesia. In addition to the inclusion criteria for the ESP block group, patients who underwent bilateral ESP block and were given local anesthetic fluid in a volume of at least 20 mL on both sides were included. In addition, patients whose local anesthetic fluid content was a mixture of 1% lidocaine + 0.25% bupivacaine were included. Patients with a local anesthetic fluid volume of less than 20 mL and a concentration of less than 1% lidocaine, or less than 0.25% bupivacaine in the content of the local anesthetic fluid mixture, were excluded from the study.

#### 2.2.2. Surgical Method and Patient Selection

Patients who underwent percutaneous balloon kyphoplasty for vertebral fracture were included in the study. Patients who underwent additional surgical procedures simultaneously as percutaneous balloon kyphoplasty were excluded from the study. The level of vertebrae that underwent surgery, the number of vertebrae that underwent surgery, the duration, and postoperative complications were recorded.

### 2.3. Comparison of Anesthesia Methods

A total of 70 patients underwent percutaneous balloon kyphoplasty, and our study was completed with the remaining 58 patients after the exclusion criteria were applied. The patients were divided into two groups according to the anesthesia method applied (Figure 1). Group GA: general anesthesia group (*n* = 19) and Group ESP: ESP block group (*n* = 39). Age, gender, ASA classification, number of surgical levels, surgical urgency, and duration of surgery of the groups were compared. It was compared with intraoperative and postoperative metrics to assess whether ESP block could be an alternative to general anesthesia. These data include whether the operation was completed with the current anesthesia method, whether there was a need for intraoperative rescue analgesia, operation time (minutes), postoperative complications, whether there was a need for postoperative ICU, number of postoperative ICU days, and number of postoperative hospitalization days.

Postoperative complications were defined as follows: hypotension: a decrease in systolic blood pressure below 90 mmHg or a decrease in systolic blood pressure of 20 mmHg from baseline; desaturation: fingertip Spo2 desaturation ≥5% from baseline; postoperative nausea and vomiting (PONV): nausea, retching, and vomiting within 24 h of surgery.

### 2.4. Outcomes

The primary outcome was to evaluate whether ESP block could be an alternative to general anesthesia as the primary anesthetic method for kyphoplasty operations. The secondary outcome was defined to compare ESP block and general anesthesia methods for kyphoplasty operation in terms of postoperative intensive care need, length of stay in the postoperative intensive care unit, and length of postoperative hospital stay.

### 2.5. Statistical Analysis

Statistical analyses were performed using IBM SPSS for Windows, Version 26.0 (IBM Corp., Armonk, NY, USA). Continuous data were expressed as mean ± SD. Categorical variables were expressed as numbers and percentages. The distribution of variables was evaluated for normality using the Kolmogorov–Smirnov test. Normally distributed data, including continuous variables, were analyzed using Student’s *t*-test for parametric tests and Mann–Whitney U test for non-parametric tests. Categorical variables were analyzed using the Chi-square test and Fisher’s exact test. If the *p* value was less than 0.05, it was considered statistically significant.

## 3. Results

The study was completed with retrospective data of 58 patients after inclusion and exclusion criteria. The patients were divided into 2 groups, 19 patients in Group GA and 39 patients in Group ESP. The median age of the patients included in the study was 64 (min-max: 23–88). Of these patients, 32 (55.2%) were female and 26 (44.8%) were male. There were 51 (87.9%) patients with ASA classification III and 7 (12.1%) with ASA classification IV. Surgery was performed urgently in 38 (65.5%) patients and electively in 20 (34.5%) patients. Vertebral levels with percutaneous balloon kyphoplasty were as follows: 1 level, 29 (50%) patients; 2 level, 21 (36.2%) patients; 3 level, 4 (6.9%) patients; and 4 level, 4 (6.9%) patients. The median duration of surgery was 57.91 (min-max: 18–102) minutes. For postoperative complications, hypotension in 6 patients (Group GA: *n* = 3, Group ESP: *n* = 3), desaturation in 7 patients (Group GA: *n* = 4, Group ESP: *n* = 3), and PONV in 1 patient (Group GA: *n* = 3, Group ESP: *n* = 3) were present. There was no statistically significant difference between Group GA and Group ESP in any of these data (Table 1 and Table 2). A total of 7 (36.6%) patients in Group GA and 6 (15.4%) patients in Group ESP needed intensive care, and there was no statistically significant difference between the groups (*p* = 0.066). However, there was a higher need for intensive care rate in Group GA (Table 1). When Group GA and Group ESP were compared in terms of intensive care unit stay (*p* = 0.025) and hospitalization stay (*p* = 0.040), a statistically significant difference was found, which was higher in Group GA. None of the patients had an incomplete operation with the current anesthesia method. None of the patients required intraoperative rescue analgesia (Table 1 and Table 2).

In addition, the vertebral levels that underwent percutaneous balloon kyphoplasty ranged from T6 to L4. The most frequently operated vertebra was L1 vertebra with 32 (32.6%) (Figure 2).

## 4. Discussion

According to the results of our study, there was no statistical difference in the data of age, gender, ASA classification, presence of emergency operation, and postoperative complications in patients who underwent general anesthesia and ESP block. In both groups, there was no operation that could not be completed with the current anesthesia method and there was no need for rescue analgesia. A statistically significant difference was found, which was higher in the general anesthesia group, when the length of intensive care stay and the length of hospital stay of the patients were compared. Although there was no statistically significant difference between the two groups regarding the need for intensive care, the need for intensive care rate was 36.8% higher in Group GA and 15.4% in Group ESP.

To detect nerve damage that may develop during percutaneous balloon kyphoplasty operations, surgeons may want patients to be awake and prefer local infiltration anesthesia to detect early nerve damage. This may lead to a decrease in patient compliance [4,5]. Patient non-compliance may cause the patient to move, thus making the operation difficult and prolonging it, and increasing the possibility of postoperative complications. For these reasons, sedo-analgesia is usually needed, although the kyphoplasty operation is performed under local infiltration anesthesia [4].

However, since kyphoplasty operations are performed in the prone position, deep sedation in the prone position is risky in terms of airway safety. Therefore, anesthetists may prefer general anesthesia with endotracheal intubation instead of sedation [5]. Nineteen of the patients in our study were in Group GA, and general anesthesia was applied to these patients with endotracheal intubation. Patients who were operated on with sedo-analgesia were excluded from the study.

General anesthesia can provide the patient with good perioperative comfort and stable hemodynamic parameters compared to local anesthesia. It also increases comfort during prone positioning for operation in patients with vertebral fractures. Since the patient will not move, the operation can be completed with a safer, shorter surgical time and less need for fluoroscopy. When general anesthesia and local infiltration anesthesia are compared, it has been reported that better results are obtained in the correction of kyphosis and in the correction of vertebral body size and configuration in patients who underwent general anesthesia [6].

However, general anesthesia may increase the incidence of postoperative complications such as PONV, hypotension, desaturation, sore throat secondary to tracheal intubation, postoperative myocardial ischemia, and lung infection [8,9,13,14,15]. In our study, hypotension, desaturation, and PONV were reported in postoperative complications. There was no statistically significant difference between Group GA and Group ESP in any of these data.

Patients with the most frequent application of balloon kyphoplasty are osteoporotic patients [1]. Osteoporosis may be associated with systemic diseases, patients may have multiple comorbidities, and the ASA classification is high [16,17,18]. Our study included 58 patients with ASA classification III and IV. Since the risk of general anesthesia is high in ASA classification III and IV patients, regional anesthetic techniques can reduce these risks [4]. ESP block is a block that can be applied to the cervical, thoracic, and lumbar regions and thus can be used in many surgeries. Although it is most commonly applied with general anesthesia and used for postoperative analgesia, its use as a primary anesthetic has also been reported [3,5]. ESP block was applied to 39 patients as the primary anesthetic method for balloon kyphoplasty surgery. Regarding the dermatomal spread of ESP block, Cassai et al. [19] reviewed 14 articles and reported that the average volume to cover a dermatome after local anesthetic injection is 3.4 mL. Additionally, in this paper, the unilateral volume of 20 mL, which was used in 11 articles, was preferred. The dermatomal spread after 20 mL of a local anesthetic to 1 side was reported as 3–8 levels, and the maximum dermatomal spread after 30 mL of local anesthetic was reported as 9 levels [19].

Chin et al. [20] reported dermatomal spread in the craniocaudal direction from the level of ESP block to the 3–6 vertebral levels. We also included patients who have applied bilaterally and at least 20 mL of a local anesthetic to 1 side in our study. According to our research data, there are patients who underwent ESP block even in multilevel percutaneous balloon kyphoplasty operations, and no return to general anesthesia was reported in any patient. According to the study by Zang et al. [13], 159 patients were included in the research, and percutaneous balloon kyphoplasty was applied to a total of 565 vertebrae. A total of 232 (41.1%) of the patients were used to T11-T12-L1-L2 levels. In our study, percutaneous balloon kyphoplasty was applied to 98 vertebrae in total, and the most frequently applied levels were L1, T12, and L2 vertebrae, respectively (Figure 2).

With the introduction of ultrasound into anesthesia practice, regional anesthetic techniques have become increasingly widespread, and many studies have been published. It has been reported to be used successfully in treating acute and chronic pain [6,10]. In addition, when compared, general anesthesia and intravenous opioids alone with regional anesthetic methods reduce opioid use, improve pulmonary outcomes, reduce adverse cardiac events, reduce mechanical ventilation time by providing earlier extubation, decrease the length of stay in the intensive care unit, decrease hospital stay, and decrease the need of intensive care. It has been reported to reduce the hospitalization rate in the care unit [21]. In our study, the rates of hospitalization in the intensive care unit, the length of stay in the intensive care unit, and the length of stay in the hospital were found to be lower in patients who underwent ESP block compared to the patients in Group GA. The patients in our study are high-risk ASA III and IV patients, and the length of hospital stay and intensive care unit stay may be due to reasons other than anesthesia. However, there was no difference between Group GA and Group ESP in terms of ASA classification. Therefore, the difference between the postoperative intensive care stay and postoperative hospital stay between the groups can be associated with the anesthesia method applied.

All surgical methods can be applied under general anesthesia. However, in the literature, comparisons of general anesthesia and other anesthesia methods have been made in many surgical methods. In order for other anesthesia methods to be considered as an alternative to general anesthesia, it can be summarized as providing adequate surgical analgesia, completing the operation with this method, and not preventing the surgeon from continuing the operation. General anesthesia with neuraxial anesthesia (spinal anesthesia, epidural anesthesia, combined spinal–epidural anesthesia, etc.), peripheral nerve blocks (infraclavicular block, interscalene block, sciatic nerve block, femoral nerve block, etc.), and trunk blocks (ESP block, paravertebral block, serratus anterior, plane block, quadratus plane block, etc.) have been compared in many studies [6,10,12,13]. In our study, general anesthesia and ESP block were compared. However, our study is the first to compare general anesthesia and ESP block for percutaneous balloon kyphoplasty operation. According to the findings we obtained in our study, between the two groups, there was no difference between the presence of an incomplete operation, the need for rescue analgesia, the duration of surgery, and postoperative complications. The data in our study suggest that ESP block may be an alternative method to general anesthesia for percutaneous balloon kyphoplasty surgery. In addition, the duration of postoperative intensive care stay and postoperative hospital stay were shorter in Group ESP. In addition, our study showed that ESP block, as a primary anesthesia method, is advantageous in terms of the need for intensive care in the postoperative period, length of intensive care stay, and length of hospital stay in high-risk patients.

## 5. Conclusions

In conclusion, ESP block may be an alternative method to general anesthesia as the primary anesthetic option for single or multilevel percutaneous balloon kyphoplasty operations. With the ESP block, the length of stay in the intensive care unit and the length of stay in the hospital can be reduced, especially in high-risk patients. These findings should be investigated in prospective randomized clinical trials.

## 6. Limitations

Our study had some limitations as it was a retrospective study. Preoperative and intraoperative pain scores, intraoperative patient comfort, and intraoperative surgeon comfort data were missing to assess whether ESP block was successful as a method of anesthesia. In addition, there were no data showing the success of the block, such as the duration of ESP block application, the number of needle insertions for the block, the control of dermatomal spread after the block, and the need for additional postoperative analgesia. The patients did not have data on admission and discharge indications to the intensive care unit in the postoperative period. In addition, there were no data on the effect of anesthesia method on surgical success. The absence of these data was a limitation of our study.

## Figures and Tables

**Figure 1 medicina-59-00240-f001:**
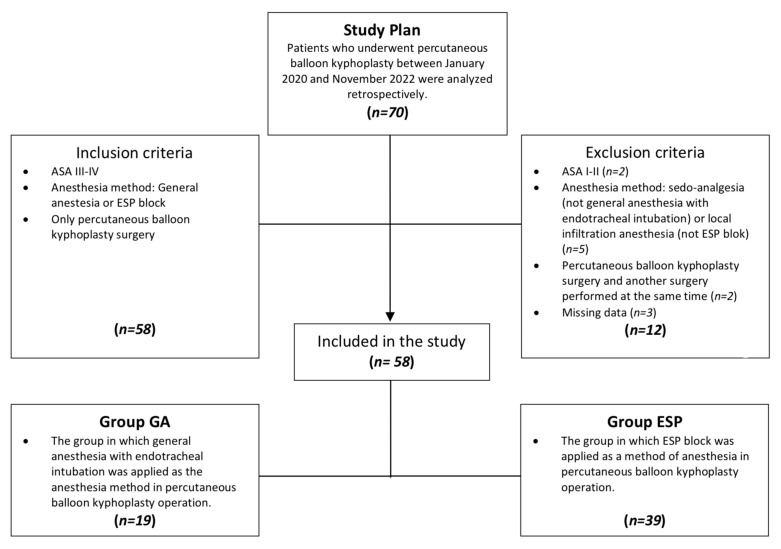
Flowchart of the current study.

**Figure 2 medicina-59-00240-f002:**
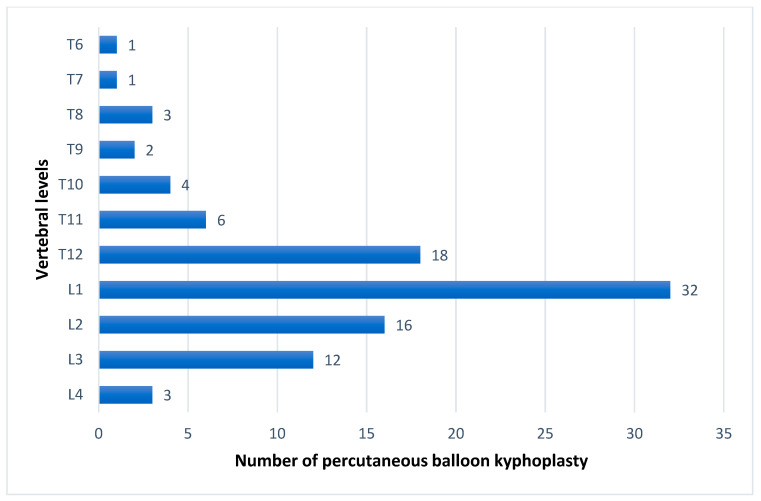
Vertebral levels and number of percutaneous balloon kyphoplasty.

**Table 1 medicina-59-00240-t001:** Data of study participants.

	All Patients (*n*: 58)	GA Group (*n*: 19)	ESP Group (*n*: 39)	*p*
Age	64 (23–88) *	58 (35–75) *	65 (23–88) *	0.344
Gender				
Male	26 (44.8%)	8 (42.1%)	18 (46.2%)	0.771
Female	32 (55.2%)	11 (57.9%)	21 (53.8%)	
ASA classification				
3	51 (87.9%)	16 (84.2%)	35 (89.7%)	0.673
4	7 (12.1%)	3 (15.8%)	4 (10.3%)	
Type of opereation				
Emergency	38 (65.5%)	14 (73.7%)	24 (61.5%)	0.361
Elective	20 (34.5%)	5 (26.3%)	15 (38.5%)	
Number of surgical levels				
1	29 (50%)	11 (57.9%)	18 (46.2%)	0.858
2	21 (36.2%)	4 (21.1%)	17 (43.6%)	
3	4 (6.9%)	2 (10.5%)	2 (5.1%)	
4	4 (6.9%)	2 (10.5%)	2 (5.1%)	

The number of patients is stated as *n* and the % values are given. *: median and min-max values. ASA: American Society of Anesthesiologists.

**Table 2 medicina-59-00240-t002:** Comparison of anesthesia methods with intraoperative and postoperative metrics.

	All Patients (*n*: 58)	GA Group (*n*: 19)	ESP Group (*n*: 39)	*p*
İncomplete operation				
Yes	0	0	0	*
No	58	19	39	
Intraoperative rescue analgesia				
Yes	0	0	0	*
No	58	19	39	
Surgical time (minute)	57.91 (18–102) **	59.88 (18–102) **	56.95 (20–78) **	0.547
Postoperative Complications				
Hypotension				
Yes	6 (10.3%)	3 (15.8%)	3 (7.7%)	0.382
No	52 (89.7%)	16 (84.2%)	36 (92.3%)	
Desaturation				
Yes	7 (12.1%)	4 (21.1%)	3 (7.7%)	0.201
No	51 (87.9%)	15 (78.9%)	36 (92.3%)	
PONV				
Yes	1 (1.7%)	1 (5.3%)	0	0.328
No	57 (98.3%)	18 (94.7%)	39 (100%)	
Postoperative ICU				
Yes	13 (22.4%)	7 (36.8%)	6 (15.4%)	0.066
No	45 (77.6%)	12 (63.2%)	33 (84.6%)	
Postoperative ICU days	0.26 (0–2) **	0.47 (0–2) **	0.15 (0–1) **	***0.025*** ***
Postoperative Hospital days	2.12 (1–6) **	2.47 (1–6) **	1.95 (1–4) **	***0.040*** ***

The number of patients is stated as *n* and the % values are given. *: no statistics are computed, **: mean and min-max values, ***: statistically significant (*p* < 0.05). PONV: postoperative nausea and vomiting, ICU: intensive care unit.

## Data Availability

All data analyzed during this study are included in the article. Further enquiries can be directed to the corresponding author.

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
