# Peer review of "Retrospective Comparison of Anesthetic Methods for Percutaneous Balloon Kyphoplasty Surgery: General Anesthesia and Erector Spinae Plane Block"

_medicina, 2023, doi:10.3390/medicina59020240_

Round 1
Reviewer 1 Report
1.In the abstract, the authors stated that “There was no statistically significant difference between Group GA and Group ESP in terms of age··· and complication data of the patients included in the study.” But the authors concluded that “With the ESP block, the length of stay in the intensive care unit, the length of stay in the hospital can be reduced, thereby reducing complications.” It is unreasonable inference! No postoperative complication was tested in the current study!
The authors should avoid making unsubstantiated assertions.
2. Ple say postoperative complication!
3. Please report the estimated treatment effect and CI here as well for each of these (for outcome). P-values alone do not tell the reader enough.
Outcome measures should be presented in separate tables
4. Please provide diagnostic criteria for postoperative complications and the specific type of complications.
5. The study is retrospective and as many covariates as possible should be collected, taking into account the identification of retrospective studies. But the study provided too little information.
6. Patients with ASA classification III or IV were included in the current study, thus, the difference in length of hospital stay after surgery may be more related to poorer basic health rather than GA or ESP block, but there is too little information included in this study to verify the reliability of the conclusions.
7. The authors stated that “The primary outcome was to evaluate whether ESP block could be an alternative to general anesthesia as the primary anesthetic method for kyphoplasty operations.” How to do it? It's hard to understand, what metrics are quantified to measure. It needs major revision!
8. Please provide the basis for the sample size and the power of the study.
9. Please provided the definition of a hospital stay. admit to hospital or postoperative hospital stay?
Reviewer 2 Report
In a retrospective study, the authors compared 2 anesthetic techniques during percutaneous balloon kyphoplasty in high-risk patients (ASA III and IV): general anesthesia and Erector Spinae Plane (ESP) block. The objective was "to investigate whether ESP block can be an alternative to general anesthesia as the primary anesthetic method for kyphoplasty operations."
The methodology used to analyze the results is inadequate. As this is a retrospective study, the authors should analyze the data on an intention-to-treat basis and not per-protocol.
Also, it is not normal to secondarily exclude patients who were converted to GA or who had received an anesthetic solution deemed insufficient. Including these patients may incidentally alter the results.
Moreover, as this is a retrospective study, the results must be presented in the conditional state.
Finally, a flow-chart should be included
Round 2
Reviewer 1 Report
Thanks for your effort.